# Spatial-Temporal Variation and Influencing Factors of Regional Tourism Carbon Emission Efficiency in China Based on Calculating Tourism Value Added

**DOI:** 10.3390/ijerph20031898

**Published:** 2023-01-19

**Authors:** Jun Liu, Fanfan Deng, Ding Wen, Qian Zhang, Ye Lin

**Affiliations:** 1School of Tourism, Hubei University, Wuhan 430062, China; 2Tourism Development and Management Research Center, Hubei Key Research Base of Humanities and Social Sciences, Wuhan 430062, China; 3South China Institute of Environmental Sciences, Ministry of Ecology and Environment, Guangzhou 510530, China; 4School of Business, Hubei University, Wuhan 430062, China

**Keywords:** tourism value added, tourism carbon emission efficiency, spatial pattern, influencing factors, geographical detector

## Abstract

Tourism-related carbon emission efficiency is an important indicator that reflects the sustainable development of tourism and can better balance the relationship between negative environmental impact and economic value. According to panel data of 30 provincial regions, “the tourism value added coefficient” (not including the Tibet Autonomous Region) in mainland China from 2000 to 2019, we estimate the tourism of each provincial administrative unit carbon emissions, measure the tourism carbon efficiency value, and analyze the measurement results of the change trend, spatial differentiation characteristics, and influencing factors. The results show that (1) the carbon emission efficiency of regional tourism in China increased significantly from 2000 to 2019, but there was a significant difference in the carbon emission efficiency of tourism among regions, and the sustainable development level of regional tourism was still unbalanced. (2) The spatial pattern of provincial administrative units in China has the adjacent characteristics of High-High agglomeration and Low-Low agglomeration, the difference in the tourism eco-efficiency development level among regions gradually decreases with time, and there is a dynamic convergence characteristic. (3) The *q* value represents the intensity of the impact factor on tourism carbon emission efficiency. According to the *q* value, the factors affecting tourism carbon emission efficiency were divided into dominant factors (0.5 ≤ *q* ≤ 1), inducing factors (0.2 ≤ *q* < 0.5) and driving factors (0 ≤ *q* < 0.2), among which the level of technological development was the dominant factor. The level of opening-up to the outside world is the inducing factor; environmental regulation intensity, urbanization level, regional economic development level, tourism industry environment, and tourism infrastructure are the driving factors. (4) The influence degree of influencing factors on the spatial differentiation of tourism carbon emission efficiency is significantly different in different periods. The degree of influence of the urbanization level and tourism industry environment shows an upward trend over time, and the influence degree of other factors shows a “V-shaped” trend. (5) The two-factor interaction will significantly enhance the spatial differentiation of regional tourism carbon emission efficiency, and the interaction between the level of scientific and technological innovation and other influencing factors has a deeper impact on tourism carbon emission efficiency.

## 1. Introduction

Tourism is not only a booster for national economic development but also an important carrier for implementing the development goals of Transforming our World: The 2030 Agenda for Sustainable Development issued by the United Nations. According to The World Tourism Economic Trend Report 2022, China’s total tourism revenue has historically ranked second in the world and ranks first among developing countries from 2019 to 2021. As a growth point of national economic development, tourism makes an important contribution to the improvement of the national economic level [1]. However, the continuous expansion of the tourism industry, tourism flow, tourist consumption behavior, and tourism product production processes will result in increased energy consumption, releasing carbon dioxide and other greenhouse gases that will have a negative impact on the environment [2,3]. In the China Energy Statistical Yearbook 2020, the total energy consumption of tourism-related transportation, warehousing and postal services, wholesale and retail trade, accommodation, and catering industries totaled 570 million tons of standard coal, accounting for 11.8% of the total annual energy consumption in 2019. According to the report released by the World Tourism Organization, carbon emissions from international tourism accounted for 8% of global greenhouse gas emissions in 2019, and carbon emissions from global tourism transportation are expected to increase to 1.998 billion tons in 2030, which makes up 5.3% of global carbon emissions.

The above data show that tourism not only creates economic benefits but also has a negative environmental impact. Tourism development should better balance the relationship between economic development and environmental impact [4,5,6]. Research on tourism carbon emissions in China has accumulated rich results, but due to the different methods and calibers, the results are quite different [7,8,9,10]. Therefore, this study will try to use the “top-down” approach to calculate tourism carbon emissions and tourism carbon emission efficiency in China from the perspective of tourism value added to better assess the sustainable development of the tourism industry in China.

## 2. Literature Review

At present, tourism carbon emissions continue to show an increasing trend [11], and tourism development has a significant enhancement effect on carbon emissions [12]. In the future, tourism may become the main source of global greenhouse gases [13], and the proposal of a “carbon peak” may challenge the development of tourism. Tourism carbon emissions have become an important issue in the field of sustainable tourism development. Some scholars [14,15] use tourism carbon emission efficiency as an important tool to measure the economic and environmental impact of tourism. The existing research on tourism carbon emission efficiency has been around since its early inception. Although some studies have not clearly proposed tourism carbon emission efficiency as a factor, their research contents are similar. At present, most of the related studies focus on the measurement of tourism carbon emission efficiency [16,17], spatial characteristics [18], and influencing factors [18,19]. Gössling [14,20] pioneered research related to this factor. In his research, he analyzed the interaction between the environmental impact and economic benefits of tourism by measuring the carbon emissions of different tourism destinations and different tourism sectors. He believed that the ecological efficiency obtained by the ratio of tourism carbon emissions to tourism income is a useful metric for analyzing the comprehensive environmental and economic performance of tourism. Sabine used a single ratio method to calculate the carbon emission intensity of the Swiss tourism industry by selecting the tourism value added and the greenhouse gas emissions of the tourism industry as indicators of economic value and environmental impact and found that the carbon emission intensity of the tourism sector is four times the average carbon emission intensity of the economic sector [21]. Moutinho used the “complete decomposition” technique to study the carbon emission intensity and its influencing factors of the five subsectors of the Portuguese tourism industry. It was found that different influencing factors have different degrees of influence on the relevant sectors. Among them, the energy over fixed capital, the capital over labor productivity, and the tourism intensity have a greater impact on the carbon emission intensity of the accommodation, food and beverage services, and the carbonization index constitutes an important effect in transport [22]. Chinese scholars’ research on tourism carbon emission efficiency and its related spatial characteristics and influencing factors has been increasing in recent years. For example, Ruan found that areas with high tourism carbon emissions were concentrated in coastal areas and economically developed areas [23]. Liu et al. compared and analyzed the tourism eco-efficiency value in China and various regions with the world’s sustainable development value and found that China’s tourism industry has entered the stage of sustainable development since 2000 [24]. Song and Li [6], Liu et al. [18], Zhang et al. [25], and others analyzed the tourism carbon emission efficiency in different regions of China using the model method and used the panel Tobit model or combined it with the σ convergence model and the β convergence model to analyze the influencing factors of tourism carbon emission efficiency or tourism eco-efficiency.

From the above research results, the key to tourism carbon emission efficiency and related research is the measurement of tourism carbon emissions. There are two main ways to estimate tourism carbon emissions. One is the “top-down” method [26,27,28,29,30,31], which is suitable for measuring tourism-related carbon emissions at the national or regional level [28], such as Lee’s [27] calculation of the tourism carbon footprint in EU countries and Avishek’s [19] calculation of tourism carbon emissions in Queensland, Australia. This method usually uses the input-output table [27,28,29] and the Tourism Satellite Account [29,30] to multiply the proportion of the value added of each sector of tourism in the value added of the sector of the national economy by the total energy consumption of the sector of the national economy, thereby obtaining tourism energy consumption and estimating tourism carbon emissions. The other is the “bottom-up” method [7,15,31], which is suitable for assessing the carbon emissions of small-scale tourist destinations [17] and medium- and large-scale tourist destinations with rich tourism-related data. For example, Chinese scholars have measured the regional tourism carbon emissions in Chinese coastal cities [18] and the Yangtze River Economic Belt [32,33]. This method is based on the perspective of the tourism sector, and tourism carbon emissions are obtained by gradually estimating the carbon emissions of each component of the tourism industry. Constrained by tourism statistics, in addition to the above two methods, research on the estimation of regional tourism carbon emissions based on the tourism consumption stripping coefficient method [34] and life cycle assessment has also been increasing in recent years [35,36,37]. In general, the measurement of tourism carbon emissions is a complex process. At present, research on this topic is in the exploratory stage, and the results of the above methods are still quite different. The main reasons for the differences are as follows. (1) Due to the differences in the development of tourism in various regions and the restrictions of research conditions, the carbon dioxide emission factor and energy consumption coefficient of transportation, accommodation facilities, and tourism activities have not been unified [14,38]. Some scholars’ research on the carbon emission factor and energy consumption coefficient used in the tourism carbon emission estimation model in recent years is still based on data from more than ten years ago, which is too old and ignores the changes in tourism caused by economic development and technological innovation [7,39]. (2) The tourism industry involves many activity sectors. The estimation results of tourism carbon emissions obtained by existing methods are often lower than the actual results, with large “leakage”.

Another key aspect of tourism carbon emission efficiency and related research is the selection of the economic value index of the tourism industry. Existing studies mainly use tourism income and tourism value added to characterize the economic value of the tourism industry. Since the statistical work of tourism value added in China has been in the exploratory stage, Chinese scholars mostly use the total tourism income to characterize the economic value of the tourism industry, and the definition and calculation of tourism value added are of great significance to objectively evaluate the status of tourism in the national economy [40]. Therefore, the calculation of tourism value added better reflects the economic contribution of tourism. At present, many scholars mainly estimate the tourism value added through the tourism satellite account method [36], tourism consumption stripping coefficient method [30], tourism income calculation method [37], etc. Anda et al. calculated the tourism value added in Romania from 2008 to 2014 through the Tourism Satellite Account. The results show that the tourism value added in Romania accounted for 2.1% of its GDP in 2014 [41]. Using the input-output table, Zeng and Cai measured the tourism value added in 30 provincial regions (not including the Tibet Autonomous Region) in mainland China in 2000 and 2005, which were 389.083 billion yuan and 860.345 billion yuan, respectively, and found that the contribution rate of the tourism industry in most regions was concentrated in 3–8% [42]. Yu took Shandong Province as the research object and measured the tourism value added by the TSA method in 2012 and 2017, attaining results of CNY 2197.35 billion and CNY 367.436 billion, respectively [43]. Wu et al. took Guangdong Province as the research object and measured that the contribution of tourism value added to Guangdong’s economy in 2019 was 2.53% [44]. Overall, the empirical research on tourism value added needs to be strengthened, especially the estimation of time series tourism value added in China, which is still relatively scarce.

In summary, scholars on tourism carbon emission efficiency and its related research are still in the stage of rapid development and present the following characteristics. First, tourism carbon emissions are a key indicator of tourism carbon emission efficiency measurement, and there is still a lack of a unified framework for the estimation of tourism carbon emissions. Second, tourism value added as a key indicator of the economic value of the tourism industry has not been widely used, and tourism income is the main indicator of current research. Third, in terms of research methods, the index method and model method are the main methods used to measure tourism carbon emissions efficiency, and the methods used to study the influencing factors are diverse. There are both regression analyses based on panel data and the geographical detector. Fourth, the research scale basically covers the country, region (large scale) to the city, and destination (small and medium scale). In view of the above research status, this study will try to use the “tourism value added coefficient” to estimate the tourism value added in China in accordance with the “top-down” estimation ideas and calculate tourism carbon emissions on this basis to further enrich the research content of tourism carbon emissions. At the same time, the geographical detector is used to study the influencing factors to further clarify the impact mechanism of tourism carbon emission efficiency.

## 3. Methods and Data Sources

### 3.1. “Tourism Value Added Coefficient” Method 

This study draws on the idea of the tourism consumption stripping coefficient proposed by Li and Li [40], estimates the tourism value added rate through tourism income and tourism value added, and then attempts to estimate tourism carbon emissions in China with the tourism value added rate. The specific calculation process is as follows.

Using the energy balance sheet of provincial regions from the *China Energy Statistical Yearbook (2001*–*2020)*, the final energy consumption of each sector in the table is converted into standard coal, and the seven sectors in the table are divided according to the primary industry, the secondary industry, and the tertiary industry. The proportion of the final energy consumption of the tertiary industry in each province in the whole $\beta_{i}$ (i = 1, 2, …, 30) is calculated, and the tertiary industry carbon emissions are obtained. Then, the tourism income is converted into tourism value added according to the tourism value added rate, and the tourism carbon emission data in each provincial region are obtained. Due to the lack of published data, it does not include the Tibet Autonomous Region.
(1)$$TC=\overset{n}{\underset{i=1}{\sum }}TC_{i}=SC_{i}\times r_{i}\;$$
(2)$$SC_{i}=C_{i}\times \beta_{i}\;$$
(3)$$r_{i}=\frac{A_{i}\times b_{i}}{T_{i}}\;$$
where *SC* is the tertiary industry carbon emissions, *C* is the total carbon emissions, and β is the estimated proportion of energy consumption of the tertiary industry. *TC* is the total tourism carbon emissions in China, *r* is the proportion of tourism value added in the added value of the tertiary industry, *A* is the tourism income, and *b* is the tourism added value rate.

### 3.2. Single Ratio Method

This paper uses a single ratio method to measure tourism carbon emission efficiency. The advantage of this method over the model method or other methods is that it is easy to use and only involves the economic indicators and environmental impact indicators of the tourism industry. In addition, this paper uses tourism value added and tourism carbon emissions to characterize the economic value index of the tourism industry and negative environmental impact, respectively. The tourism carbon emission efficiency obtained by this method can be compared with that of other industries to evaluate the sustainable development level of tourism more scientifically. The specific calculation method is as follows (4), where *TVA* is the tourism value added and *TCE* is the tourism carbon emission efficiency.
(4)$$TCE_{i}=\frac{TC_{i}}{TVA_{i}}$$

### 3.3. Exploratory Spatial Data Analysis (ESDA)

Analysis of spatial patterns can test the spatial relationship of tourism carbon emission efficiency. Global spatial autocorrelation is generally measured by Moran’s I index, which is used to explain whether there is a significant correlation between spatial elements. Therefore, this study will use global spatial autocorrelation to analyze the spatial pattern characteristics of tourism carbon emission efficiency. Moran’s I is calculated using Formula (5):(5)$$Moran^{\prime }s\;I=\frac{n\sum_{i=1}^{n}\sum_{j=1}^{n}w_{ij}\left(x_{i}-\bar{x}\right)\left(x_{j}-\bar{x}\right)}{(\sum_{i=1}^{n}\sum_{j=1}^{n}w_{ij})\sum_{i=1}^{n}{\left(x_{i}-\bar{x}\right)}^{2}}$$
where *x* is the tourism carbon emission efficiency, *W* is the spatial weight matrix, and the value range of *I* is [−1, 1]. When the value of *I* is (0, 1], the tourism carbon emission efficiency is positively correlated; when the value of *I* is [−1, 0), tourism carbon emission efficiency is negatively correlated; and when *I* = 0, there is no spatial relationship between tourism carbon emission efficiency.

### 3.4. Geographical Detector 

A geographical detector is one of the main tools for driving force and factor analysis [45]. The core idea is based on the following hypothesis. If an independent variable has an important influence on a dependent variable, then the spatial distribution of independent variables and dependent variables should be similar. The geographic detector includes four detectors: a factor detector, interaction detector, ecological detector, and risk detector, which can detect both numerical data and qualitative data. This method calculates and compares the *q* value of every single factor and the *q* value after the superposition of the two factors to determine whether there is an interaction between the two factors, as well as the strength, direction, linearity, or nonlinearity of the interaction. The calculation formula of the *q* value is as follows:(6)$$q=1-\frac{\sum_{h=1}^{l}n_{h}\sigma_{h}^{2}}{n\sigma^{2}}$$
where *h* is the stratification of the dependent variable *Y* or the dependent variable *X*, that is, classification or partition; $n_{h}$ and n are the number of elements in layer *h* and the whole region, respectively; and $\sigma_{h}^{2}$ and $\sigma^{2}$ are the variances of the layer h and the region *Y* values. The value range of *q* is [0, 1]. The larger the value is, the stronger the explanatory power of the impact factor on the carbon emission efficiency of tourism. The value of 0 indicates that the impact factor is completely independent of the carbon emission efficiency of tourism. The value of 1 indicates that the impact factor can fully explain the distribution difference in the carbon emission efficiency of tourism. Since the geographical detector can only process type variables, it is necessary to use the classification algorithm to discretize the numerical variables.

### 3.5. Influencing Factors Selection and Framework Construction

From the measurement results and differences analysis of tourism carbon emission efficiency, we find that there are significant differences in the sustainable development level of regional tourism in China, which indicates that tourism carbon emission efficiency in various provincial regions may be affected by many factors. Considering the difficulty of data acquisition, comprehensiveness of data, and limitations of the measurement model, it is unlikely to take all influencing factors into consideration. Therefore, this study mainly selects the influencing factors of tourism carbon emission efficiency based on the following aspects. (1) Based on the measurement model of tourism carbon emission efficiency, this study explores the influence mechanism of various factors on tourism carbon emission efficiency from the perspectives of regional development, tourism industry development, and resource environment. (2) Referring to the existing research results on tourism carbon emission efficiency, we select the level of economic development, the level of opening-up, the level of urbanization, the tourism industry environment, the tourism infrastructure, the level of scientific and technological innovation, and the intensity of environmental regulation. Based on this, this paper constructs the influencing factors framework of China’s regional tourism carbon emission efficiency (Figure 1).

(1)The level of economic development: the level of economic development affects capital investment, resource consumption, industrial structure, and technology research and development and upgrading through the scale effect, structure effect, and technological effect [46], which in turn affect the input and output of tourism carbon emission efficiency.(2)The level of opening-up: the higher the level of opening-up to the outside world, the higher the total amount of foreign investment that the tourism industry can introduce, and the more advanced the introduction of eco-tourism management experience and production equipment, the more it can provide new opportunities for the economic development of the tourism industry and new technologies for environmental governance.(3)The level of urbanization: with the gradual increase in the urban population, talents and various elements in various regions continue to gather, and the level of human capital is further improved, driven by the desire for exotic travel experiences and an increasing reliance on aviation and luxury amenities, affluence has turned tourism into a carbon-intensive consumption category [47,48], thus affecting resource consumption and environmental governance.(4)Tourism industry environment: this study uses tourism resource endowment (TRE) as a measure of the tourism industry environment to explore its impact on tourism carbon emission efficiency. Tourism resources provide strong support for the development of the tourism industry and are the most important factor in attracting tourists to destinations [49]. The richer the tourism resources, the development of tourism economy is slow [46].(5)Tourism infrastructure: transportation infrastructure is an indispensable material basis for the development of the tourism industry. China’s tourism carbon emission calculation data show that the carbon emission from transportation is the main source of tourism carbon emission in China [38].(6)The level of scientific and technological innovation [48]: the continuous progress of scientific and technological innovation has two impacts on tourism carbon emission efficiency. First, technological progress can promote the modernization of labor means, achieve cleaner production and improve resource efficiency. Second, technological progress improves the level of pollutant treatment, recovery, and recycling, reduces pollutant emissions, and improves environmental efficiency.(7)The intensity of environmental regulation: environmental resources are public goods and the market is irrational. The environmental problems caused by environmental resources need to be regulated by the government using environmental policy tools. Environmental policy acts on the resource–environment system through the innovation incentive effect and environmental awareness effect, and promotes economic growth, promotes technological innovation, strengthens social supervision, improves environmental awareness, and achieves a “win-win” situation for economic benefits and the resource environment to achieve the goal of sustainable tourism development [50].

The illustration of the influencing factor indices of tourism carbon emissions efficiency is shown in Table 1.

Some indicators are calculated as follows:

The tourism resource endowment index refers to the existing research results [46,51], and combined with the actual differences of different grades of scenic spots and star hotels, it assigns five points, three points, and one point to the number of AAAAA, AAAA, and AAA scenic spots, and generates the “tourism scenic spot resources index”. The number of five-star hotels, four-star hotels, and three-star hotels is assigned according to five points, three points, and one point, respectively, to obtain the “star hotel resource index”, and then the tourism resource endowment index is summarized.

Tourism transportation infrastructure is weighted and calculated by different levels of road transportation mileage [52], and the specific calculation formula is as follows:$$Trans=\overset{n}{\underset{i=1}{\sum }}\left(\frac{V_{i}L_{i}}{100S}\right)$$

In this study, *n* is 5, $V_{i}\;$is the speed of roads of type *i* (120 km/h for railways, 100 km/h for expressways, 80 km/h for primary highways, 60 km/h for secondary highways, and 40 km/h for grade highways), $L_{i}\;$is the length of road transport lines of type *i*, and *S* is the area of the province.

### 3.6. Data Source

Considering data availability, this study uses 30 provincial regions in China (not including Hong Kong, Macao, Taiwan, and the Tibet Autonomous Region) as the object to measure the tourism carbon emission efficiency from 2000 to 2019 and selects the index data from the Yearbook of China Tourism Statistics, the China Statistical Yearbook, the China Energy Statistics Yearbook, the China Statistics Yearbook on Environment, the provincial Statistical Yearbooks, and the provincial Statistical Bulletin of Social and Economic Development. Otherwise, the total carbon emissions in China come from the China Carbon Accounting Database (https://www.ceads.net.cn/ (accessed on 5 June 2022)), and the regional marketization index comes from the China Marketization Index Database (https://cmi.ssap.com.cn/ (accessed on 5 June 2022)). The tourism value added rate is collected through local public news reports, and the interpolation method is used to supplement the areas lacking the tourism value added rate.

## 4. Results

### 4.1. Measurement Results of Tourism Value Added and Tourism Carbon Emissions

The tourism value added and tourism carbon emissions in each provincial region of China from 2000 to 2019 are estimated according to Formulas (1)–(3). Table 2 shows the measurement results of provincial tourism carbon emissions and tourism value added in 2000, 2005, 2010, 2015, and 2019. We find that the tourism value added in each provincial region shows a significant upward trend from 2000 to 2019, and the gap between the tourism value added in different provincial regions gradually narrows. The tourism value added in Guangdong, Jiangsu, and Shandong ranks in the top three, and the development of tourism is in a leading position in the country. In 2019, the contributions of tourism to regional economic development were 6.94%, 6.43%, and 8.08%, respectively. The tourism value added in Ningxia, Qinghai, and Gansu is relatively low, but its impact on the local economy cannot be ignored. In 2019, the tourism value added in the above regions accounted for 2.75%, 5.00%, and 7.01% of the regional GDP, respectively.

From the measurement results of tourism carbon emissions, we find that the tourism carbon emissions in each region of China increased from 6.78 to 513.33 million tons to 112.51 to 5873.52 million tons, with an average annual growth rate of 1.77–24.44% from 2000 to 2019, and there are significant differences between provincial regions. Among them, the tourism carbon emissions and their rising trend in Inner Mongolia, Shanxi, Gansu, Guizhou, and Hunan are significantly higher than those in other regions. The tourism carbon emissions in Beijing, Shanghai, Zhejiang, and other places showed a downward trend, and the tourism carbon emissions in Guangdong, Jiangsu, Hubei, and Liaoning growth gradually slowed down. The above results show that the tourism industry in provincial regions with higher levels of economic development may take the lead in entering the carbon peak stage.

Figure 2 shows the trend of tourism carbon emissions in China from 2000 to 2019. The tourism carbon emissions in China showed a significant upward trend, with an average annual growth rate of 13.09%, which is basically consistent with the development trend of China’s tourism industry. In 2003, due to SARS, the development of tourism was seriously hindered, and the growth rate of tourism carbon emissions showed a negative value, while the growth rate of tourism carbon emissions in other years was positive. In 2000, the tourism carbon emissions in China were 0.59 billion tons, accounting for 1.93% of China’s total carbon emissions. The tourism carbon emissions in China amounted to 609 million tons in 2019, accounting for 5.60% of China’s total carbon emissions, which is lower than the share of global tourism carbon emissions. The growth rates of tourism carbon emissions are 15.93%, 15.07%, 9.75%, and 12.71% from the Tenth Five-Year Plan to the Thirteenth Five-Year Plan, respectively. The growth rate of tourism carbon emissions in China has slowed to some extent, which is related to the fact that China has incorporated carbon emissions into the binding targets of national economic development since the Eleventh Five-Year Plan.

### 4.2. Measurement Results of Tourism Carbon Emission Efficiency

The tourism carbon emission efficiency factor can better balance the relationship between environmental impact and economic value. The single index method is used to measure the tourism carbon emission efficiency in 30 provincial regions (not including the Tibet Autonomous Region) in mainland China from 2000 to 2019. The results in the Table 3 are as follows (due to space limitations, only carbon emission efficiency results for tourism in key years are presented). (1) The overall average value of tourism carbon emission efficiency in China decreased from 0.253 kgCO_2_-e/CNY in 2000 to 0.081 kgCO_2_-e/CNY in 2019, which indicates that the overall level of the sustainable development of the tourism industry in China has maintained an upward trend, and the carbon emissions generated by the national creation of unit tourism value added have gradually decreased. During this period, the proportion of coal in the total energy consumption in China decreased significantly, from 68.5% in 2000 to 57.7% in 2019. The change in the energy consumption structure significantly improves tourism carbon emission efficiency. At the same time, tourism carbon emission efficiency dropped by 61.42% in 2019 compared with 2005, achieving China’s goal of reducing carbon emissions per unit of GDP by 40% to 45% by 2020 compared to 2005. (2) From the perspective of specific provincial regions, the value of tourism carbon emission efficiency in all regions shows a downward trend, with Guizhou, Tianjin, Beijing, Sichuan, Yunnan, Hebei, and Zhejiang falling by more than 75% and Inner Mongolia, Shandong, Heilongjiang, and Hunan falling by less than 50%. (3) From the perspective of the sustainable development level of tourism, compared with the world sustainable development threshold of 0.0334 kgCO_2_-e/CNY (referring to the studies of Gössling et al. (2005) and Liu et al. (2018), the conversion is obtained), only Beijing entered the state of sustainable development in 2019, Shanghai, Jiangsu, and Zhejiang were close to entering the state of sustainable development, and the rest of the region had a certain gap from the level of sustainable development. This shows that economically developed regions may have more advantages in energy conservation and emission reduction, scientific and technological innovation, and so on, so the above regions will take the lead in entering the state of sustainable development. (4) Compared with other industries, China’s overall average carbon emission efficiency in 2005, 2010, 2015, and 2019 was 0.29 kgCO_2_-e/CNY, 0.19 kgCO_2_-e/CNY, 0.13 kgCO_2_-e/CNY, and 0.10 kgCO_2_-e/CNY, respectively, which was higher than that of tourism. The tourism carbon emission efficiency is better than the average level of China. Therefore, the development of tourism for regional emission reduction effects is more obvious. Giving priority to the development of tourism is one of the feasible paths of carbon peak and carbon neutralization in various regions.

### 4.3. Spatial Autocorrelation Analysis

Table 4 shows the global Moran’s I index of tourism carbon emission efficiency in China from 2000 to 2019. According to the significance results, the global Moran’s I index of tourism carbon emission efficiency in China passed the significance test at the 10% level in 2000, passed the significance test at the 5% level in 2001, 2002, 2003, and 2016, and passed the significance test at the 1% level in the other years. This shows that there is spatial agglomeration in the tourism carbon emission efficiency in China. That is, the spatial distribution of tourism carbon emission efficiency in various regions presents the adjacent characteristics of High-High clusters and Low-Low clusters. However, the trend of the global Moran’s I index indicates that the correlation between tourism carbon emission efficiency and spatial distribution fluctuates. Due to the obvious gradient of China’s economic development, the tourism industry in the eastern region developed earlier and has a better foundation than that in the central and western regions. The tourism industry in parts of the central and western regions starts late but develops rapidly. The tourism value added and tourism carbon emissions have increased significantly, showing the volatility of spatial autocorrelation, which causes the tourism carbon emission efficiency also show volatility.

According to the measurement results, the tourism carbon emission efficiency value in China was lower than 1 from 2000 to 2019. We divide the tourism carbon emission efficiency of 30 provincial regions into the lowest level (0.4~1.00), lower level (0.30~0.40), middle level (0.20~0.30), higher level (0.10~0.20), and highest level (0~0.10). To further identify the High-High (the regions with low tourism carbon emission efficiency values are surrounded by the regions with low tourism carbon emission efficiency values) clusters, Low-Low (the regions with high tourism carbon emission efficiency values are surrounded by the regions with high tourism carbon emission efficiency values) clusters, Low-High (the regions with high tourism carbon emission efficiency values are surrounded by the regions with low tourism carbon emission efficiency values) clusters, and High-Low (the regions with low tourism carbon emission efficiency values are surrounded by the regions with high tourism carbon emission efficiency values) cluster characteristics of the tourism carbon emission efficiency, we conducted a local spatial autocorrelation analysis.

Figure 3 presents a Local Indicators of Spatial Association (LISA) cluster diagram of the tourism carbon emission efficiency in 2000, 2005, 2010, 2015, and 2019, and the results are all significant at the 1% level. From the perspective of development level, except for Inner Mongolia, the tourism carbon emission efficiency in most parts of the regions of China entered the highest level or higher level in 2019, 13 regions jumped from the lowest level, lower level, and middle level to the highest level or higher level, and the tourism carbon emission efficiency improved significantly. From the perspective of China’s regional patterns, the southeast coastal areas have always been in the “leading region” of tourism carbon emission efficiency, which may be due to the good foundation of the abovementioned regional economic development level and science and technology innovation level. As an important part of the regional economy, tourism enjoys regional technology spillover, which enables it to generate fewer carbon emissions per unit of tourism added value. Except for Guizhou, all regions in the southern region entered the highest level of tourism carbon emission efficiency in 2015, which was significantly better than that in the northern region. This may be related to the fact that the proportion of coal consumption in the energy consumption structure of the southern region is generally lower than that of the northern region, but the proportion of coal consumption in Guizhou has been at a higher level, with the proportion of coal consumption still as high as 69% in 2019. From the perspective of local clusters, Shanghai, Jiangsu, Zhejiang, Anhui, Fujian, and Jiangxi always showed High-High clusters. The above regions are closely linked economically. The technology spillover of Jiangsu, Zhejiang, and Shanghai is easily transmitted to Anhui, Jiangxi, and Fujian, which makes the tourism carbon emission efficiency show significant local cluster characteristics. Because Hubei and Hunan are located in the Yangtze River Economic Belt, they are also in High-High clusters most of the time. However, because Hubei and Hunan are in the transition zone from the eastern region to the western region, their local cluster characteristics are unstable. Guizhou has always shown Low-High clusters, indicating that the tourism carbon emission efficiency in the surrounding regions is better than that of Guizhou. The main reason is that its energy consumption structure is dominated by coal, and the tourism value added is significantly lower than that of the surrounding regions. In 2015 and 2019, Jilin showed significant Low-Low clusters, indicating that the tourism carbon emission efficiency in Jilin and surrounding regions was higher.

### 4.4. Estimation of Influencing Factors

#### 4.4.1. Influence of Individual Explanatory Variables

The spatial difference in tourism carbon emission efficiency in China is influenced by many factors. To identify the driving factors and their interaction on provincial tourism carbon emission efficiency in the three different periods of the Eleventh Five-Year Plan, the Twelfth Five-Year Plan, and the Thirteenth Five-Year Plan in China, the geographic detector was used in this section. The independent variables are divided into five categories by the natural discontinuity point method of ArcGIS 10.8, and then the action intensity value (*q*-value) of each index is calculated by the geographic detector mode. The greater the *q* value is, the greater the action intensity, and vice versa. According to the *q*-value, the various indicators that affect the spatial differentiation of tourism carbon emission efficiency are divided into dominant factors (0.5 ≤ *q* ≤ 1), inducing factors (0.2 ≤ *q* < 0.5), and driving factors (0 ≤ *q* < 0.2).

Table 5 reports the results of the factor detector module. In addition to the tourism industry environment in the Twelfth Five-Year Plan not passing the significance testing, the remaining factors in different periods passed the significance test at the 5% level. Different influencing factors have different impacts on the spatial differentiation of tourism carbon emission efficiency in China. The *q*-values illustrate that the explanatory power of the seven driving factors above on tourism carbon emission efficiency ranged from 0.0656 to 0.5081 from 2005 to 2019. The *q*-value of the scientific and technological innovation level is always the largest among the selected impact factors, which is the dominant factor affecting the spatial differentiation of tourism carbon emission efficiency. On the one hand, the level of scientific and technological innovation changes the energy consumption structure and reduces pollutant emissions through the improvement of the technological level; on the other hand, it improves the efficiency of resource utilization by improving the capacity of pollutant treatment and jointly promotes the sustainable development of tourism. The *q*-value of the level of opening-up is always greater than 0.2 in different periods, which is the inducing factor of the spatial differentiation of tourism carbon emission efficiency. The technical effect of the level of opening-up is reflected in the technological spillover of developed regions, which provides technical support and advanced equipment for energy conservation and emission reduction of local tourism. The structural effect promotes the high-quality development of tourism. A higher level of opening-up means more tourism foreign exchange income, which makes the tourism value added created by unit carbon emissions higher. The influence of the environmental regulation intensity, the level of urbanization, the level of regional economic development, tourism infrastructure, and tourism industry environment are weak, which is the driving factor affecting the tourism carbon emission efficiency.

The influence of the same factors on the spatial differentiation of tourism carbon emission efficiency in different periods is also changing. The influence degree of the level of urbanization and tourism infrastructure is on the rise over time, and the influence degree of other factors is in a “V” shape. The influence of the level of regional economic development and tourism infrastructure on the spatial differentiation of tourism carbon emission efficiency in different periods is more significant, which is less than 0.2 during the Eleventh Five-Year Plan and the Twelfth Five-Year Plan, while during the Thirteenth Five-Year Plan, its impact increased significantly. This indicates that during the Thirteenth Five-Year Plan, the high-quality changes in regional economic development increased the gap between tourism infrastructure construction in different regions, which in turn increased the impact on tourism carbon emission efficiency. The level of urbanization enhances the level of scientific and technological innovation in the region and increases the impact on tourism carbon emission efficiency by gathering talent and technologies for regional low-carbon tourism and talent introduction.

#### 4.4.2. Interactive Influence of Explanatory Variable Pairs

We can compare the influence of individual explanatory variables and the interactive influence of explanatory variable pairs on tourism carbon emission efficiency from the results of the interaction detector module and then judge the type of two-factor interaction. Table 6 shows the interaction results of seven influencing factors on the spatial differentiation of tourism carbon emission efficiency in different periods. The diagonal in the table is the influence degree of each factor on the spatial differentiation of tourism carbon emission efficiency, and the rest is the influence degree of the interaction between the two factors on the spatial differentiation of tourism carbon emission efficiency. Compared with the single factor influence degree value (*q*-value), the influence degree of the two-factor interaction on the spatial differentiation of tourism carbon emission efficiency is greater than that of the single factor, which indicates that the two-factor interaction can enhance the spatial differentiation of tourism carbon emission efficiency; that is, it is manifested as nonlinear enhancement q(X_i_∩X_j_) > q(Xi) + q(X_j_) or two-factor enhancement q(X_i_ ∩ X_j_) > max{q(X_i_), q(X_j_)}.

The interactive influence of explanatory variable pairs on tourism carbon emission efficiency is significantly greater than that of individual explanatory variables, and the type of interaction enhancement is obvious. During the Eleventh Five-Year Plan, the interaction intensity between the level of scientific and technological innovation and the level of urbanization reached 0.8257, which is greater than the *q*-value of the level of scientific and technological innovation (0.6729) and the *q*-value of the level of urbanization (0.1164), which belongs to nonlinear enhancement. This shows that the interactive influence of explanatory variable pairs of scientific and technological innovation level and urbanization level can explain 82.57% of the spatial differentiation of tourism carbon emission efficiency. Both the level of technological innovation and urbanization can aggravate their impact on the spatial differentiation of tourism carbon emission efficiency through technical effects. During the Twelfth Five-Year Plan and the Thirteenth Five-Year Plan, the strongest interactive influence of explanatory variable pairs are the level of scientific and technological innovation and the intensity of environmental regulation, and both show nonlinear enhancement. This indicates that the resource environment is the main factor affecting tourism carbon emissions efficiency. The level of economic development and the level of urbanization in different periods are the weakest, showing a two-factor enhancement. One possible reason is that regional economic development provides a boost for the sustainable development of tourism, and it will continue to promote the level of urbanization. The improvement of the urbanization level will increase the demand of tourists for tourism resource endowment and tourism infrastructure, making the environmental carrying capacity gradually unable to meet the increasing demand of tourists. The pressure of low-carbon tourism development is increasing, and its production–consumption effect will reduce the sustainable development level of tourism. Therefore, when the level of regional economic development and the level of urbanization work together on tourism carbon emission efficiency, the interaction intensity is weak. The interactive influence of explanatory variable pairs shows both nonlinear enhancement and two-factor enhancement. However, the interactive influence of explanatory variable pairs is obviously greater than the influence of individual explanatory variables on the spatial differentiation of regional tourism carbon emission efficiency, indicating that any factor can promote the influence of other factors on the spatial differentiation of regional tourism carbon emission efficiency. The interactive influence of explanatory variable pairs will significantly enhance the spatial differentiation of tourism carbon emission efficiency. Therefore, in the process of promoting the sustainable development of tourism, the influencing factors should be considered comprehensively, and the combination of multiple factors will be more conducive to the improvement of tourism carbon emission efficiency.

## 5. Conclusions and Discussion

Based on Chinese provincial data, we conducted an empirical study on tourism carbon emission efficiency, analyzing its regional differences, spatial autocorrelation, and influencing factors. The main conclusions of this study are as follows. First, from 2000 to 2019, the tourism value added in each region showed a significant upward trend. The tourism development of Guangdong, Jiangsu, and Shandong has always been in a leading position in the provincial regions; the tourism value added in Ningxia, Qinghai, Gansu, and other places is lower, but the impact on regional development remains significant. The tourism carbon emissions in China showed a significant upward trend, with an average annual growth rate of 13.09%; the proportion of tourism carbon emissions in China’s total carbon emissions continues to increase, from 1.93% in 2000 to 5.60% in 2019. The overall average value of tourism carbon emission efficiency in China decreased from 0.253 kgCO_2_-e/CNY in 2000 to 0.081 kgCO_2_-e/CNY in 2019, but there was a significant gap in carbon emission efficiency between provincial regions. This shows that the level of sustainable development of China’s regional tourism has been optimized, but the level of sustainable development of interprovincial tourism is still uneven. Second, each region shows a significant positive spatial correlation in the spatial pattern, there are local High-High clusters and Low-Low clusters in the spatial distribution, and the degree of global spatial agglomeration fluctuates over time. The number of regions with a high level of tourism carbon emission efficiency is increasing, the degree of difference in the sustainable development level of tourism among regions is gradually decreasing, the polarization phenomenon is weakening, and there is a dynamic convergence feature. Third, the factors that affect the spatial differentiation of tourism carbon emission efficiency are analyzed by a geographical detector. According to the *q*-value, all types of indices are divided into dominant factors (0.5 ≤ *q* ≤ 1), inducing factors (0.2 ≤ *q* < 0.5), and driving factors (0 ≤ *q* < 0.2). The level of scientific and technological innovation is the dominant factor, and tourism infrastructure, regional economic development, the level of opening-up, and the level of urbanization all have technical effects, jointly promoting the level of scientific and technological innovation and exacerbating its impact on the spatial differentiation of tourism carbon emission efficiency. The level of opening-up is the inducing factor; the intensity of environmental regulation, the level of urbanization, the level of economic development, the tourism industry environment, and tourism infrastructure are the driving factors affecting tourism carbon emission efficiency. The degree of influence of the same influencing factor on the spatial differentiation of tourism carbon emission efficiency is also significantly different during separate periods. The influence of the urbanization level and tourism industry environment shows an upward trend with time, and the influence degree of other factors shows a “V-shaped” trend. Fourth, the results of factor interaction show that the interactive influence of explanatory variable pairs on the spatial differentiation of tourism carbon emission efficiency is significantly greater than that of a single factor, which shows nonlinear enhancement and double factor enhancement, respectively, indicating that the interaction between two factors will significantly enhance the spatial differentiation of tourism carbon emission efficiency, among which the interaction between the level of scientific and technological innovation and other influencing factors has a deeper impact on tourism carbon emission efficiency.

For all regions, only Beijing’s tourism industry entered a state of sustainable development in 2019, so it is necessary to continuously improve the tourism carbon emission efficiency in all regions. Based on the above conclusion, we propose the following recommendations. First, the energy consumption structure should be optimized. From the regional differences in tourism carbon emission efficiency, it can be found that the values of tourism carbon emission efficiency in regions with high traditional coal energy consumption (the provinces in China with a high proportion of regional energy consumption structure in 2019 include Hebei, Ningxia, Shanxi, Inner Mongolia, Yunnan, and Guizhou) are higher than those in other regions. Therefore, continuously optimizing the regional energy consumption structure and accelerating the development of natural gas, actively developing hydropower, developing renewable energy sources such as wind energy, solar energy, water energy, and biomass energy, and gradually increasing the proportion of nuclear energy, renewable energy, and new energy will promote tourism carbon emission efficiency. Second, enhance scientific and technological innovation capabilities. Scientific and technological innovation promotes the low-carbon and green development of tourism by acting on all aspects of the technological effect, and scientific and technological innovation is the dominant factor influencing tourism carbon emission efficiency. Therefore, we can support scientific and technological innovation, strengthen scientific and technological support for tourism, and promote the improvement of tourism carbon emission efficiency by breaking through new efficient and clean utilization technology of coal, rail transit technology, pure electric vehicle technology, solar and photovoltaic power generation technology, wind power generation technology, etc. Third, the opening of the tourism market should be accelerated. The high-quality development of tourism requires the construction of a domestic and international dual-cycle market. By expanding market opening, tourism development factors such as capital flow, people flow, logistics, and technology flow will accelerate agglomeration, the tourism consumption structure will continue to optimize, and tourism carbon emission efficiency will also increase. Fourth, consumption-side carbon emission reduction should be strengthened. By using means (such as markets, prices, etc.) to constrain the expansion of energy consumption and encourage tourists to actively practice the concept of low-carbon tourism at the consumer end, reduce per capita tourism carbon emissions, and reduce the total amount of tourism carbon emissions from the demand side.

The index method is used to construct the tourism carbon emission efficiency measurement model in China in this study. Tourism carbon emissions are used to characterize the environmental influencing factors, and tourism value added is used to characterize the economic value index of the tourism industry, which can better describe the level of sustainable tourism development in China to some extent. Due to data limitations, this paper assumes that the tourism value added coefficient of each provincial region in China is always a constant value, which may affect the accuracy of tourism value added estimates to some extent. With the improvement of China’s tourism statistics, the estimation of tourism value added can be further improved and refined in subsequent studies. At the same time, the estimation of tourism value added also shows that the total tourism added value of each provincial region is greater than the added value of tourism and related industries announced by the National Bureau of Statistics in China from 2017 to 2019. However, considering that the tourism income of each provincial region in China is much larger than the tourism income data published by the state, the estimation of this paper is still reasonable.

## Figures and Tables

**Figure 1 ijerph-20-01898-f001:**
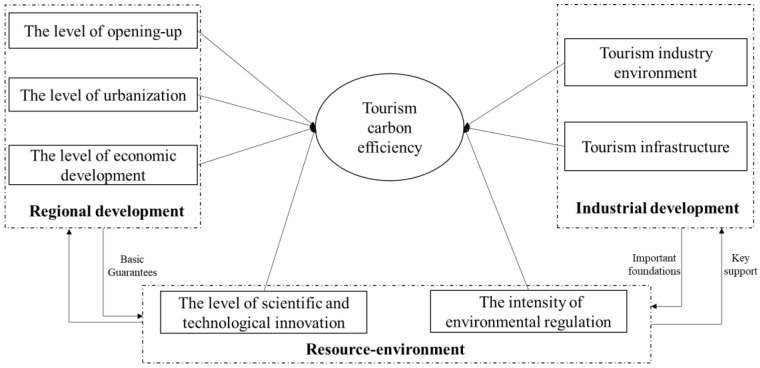
Framework of the influencing factor of tourism carbon emissions efficiency.

**Figure 2 ijerph-20-01898-f002:**
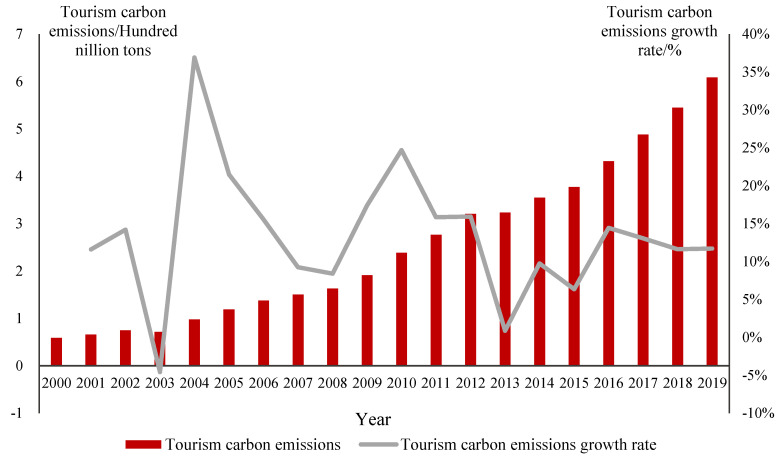
Tourism carbon emissions trends from 2000 to 2019.

**Figure 3 ijerph-20-01898-f003:**
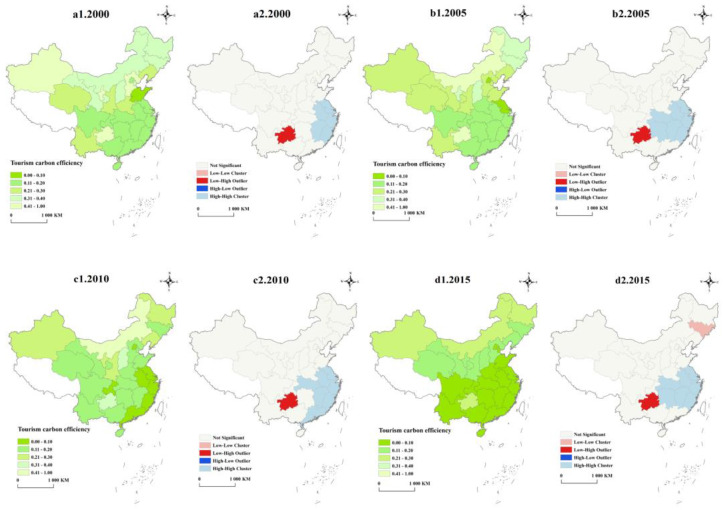
Spatial pattern and LISA cluster of tourism carbon emissions efficiency.

**Table 1 ijerph-20-01898-t001:** Illustration of the influencing factor indices of tourism carbon emissions efficiency.

Metric Type	Influencing Factors	Indicator Description	Variable Symbols
Regional development	The level of economic development	Per capita gross regional product	GRP
	The level of opening-up	Marketization index	MIN
	Level of urbanization	Urbanization rate	UR
Development of the tourism industry	Tourism industry environment	Tourism resources endowment	TRE
	Tourism infrastructure	Transportation infrastructure	TTI
Resource-environment	The level of scientific and technological innovation	Energy consumption per CNY 10,000 GDP	ECP
	The intensity of environmental regulation	Total investment in environmental pollution control	IEP

**Table 2 ijerph-20-01898-t002:** The results of tourism value added and tourism carbon emissions from 2000 to 2019.

Region	Tourism Carbon Emissions/Million Tons					Tourism Value Added/Ten Billion Yuan				
	2000	2005	2010	2015	2019	2000	2005	2010	2015	2019
Beijing	4.12	5.77	6.06	6.32	5.74	3.63	6.34	11.01	18.34	24.77
Tianjin	4.31	4.58	5.50	7.39	9.40	1.38	2.28	4.92	10.74	17.01
Hebei	3.46	3.72	6.01	17.39	35.23	0.83	1.70	3.56	13.36	36.23
Shanxi	1.63	5.36	20.87	34.80	58.74	0.44	1.56	5.81	18.48	43.03
Inner Mongolia	0.63	4.25	17.19	28.14	40.01	0.18	0.89	3.15	9.67	19.95
Liaoning	2.15	6.24	20.51	18.69	25.02	0.91	2.63	9.62	13.34	22.28
Jilin	0.87	3.42	6.04	13.56	23.84	0.26	1.05	3.36	10.38	22.54
Heilongjiang	1.18	2.22	6.28	8.90	12.95	0.35	0.72	2.28	3.51	6.93
Shanghai	5.13	8.80	9.96	6.88	7.73	3.81	6.69	12.39	14.15	22.49
Jiangsu	3.01	7.31	12.34	16.85	22.04	2.89	8.07	20.57	40.09	63.46
Zhejiang	2.52	5.94	10.20	14.79	13.85	1.73	5.05	12.12	26.14	39.93
Anhui	1.03	1.52	5.27	17.79	27.14	0.74	1.43	5.37	19.37	40.07
Fujian	1.34	2.97	4.21	5.88	11.68	1.12	2.50	5.14	11.51	29.65
Jiangxi	1.18	2.41	4.27	13.84	30.67	0.65	1.54	3.94	17.52	46.54
Shandong	2.16	12.09	23.99	24.22	33.46	2.08	5.22	15.40	33.63	56.95
Henan	3.84	6.58	12.06	19.55	26.09	1.60	3.58	10.41	22.68	42.83
Hubei	2.30	3.39	9.12	13.42	18.33	1.21	2.03	6.25	18.45	29.65
Hunan	0.45	1.64	3.92	7.34	15.32	0.37	1.13	3.57	9.28	24.42
Guangdong	7.80	10.66	14.05	22.82	29.33	5.71	9.28	19.10	45.09	74.97
Guangxi	0.97	1.89	4.71	9.14	22.17	0.66	1.18	3.67	12.53	39.43
Hainan	0.73	1.03	1.66	2.67	3.08	0.42	0.66	1.37	2.89	5.63
Chongqing	0.80	1.64	3.14	6.59	10.95	0.63	1.28	3.89	9.96	25.11
Sichuan	1.81	4.18	7.95	13.42	18.60	0.92	2.57	6.71	22.11	41.27
Guizhou	0.90	2.04	5.71	12.59	29.29	0.10	0.38	1.60	5.31	18.60
Yunnan	1.18	2.53	4.14	6.63	14.15	0.58	1.18	2.78	9.06	30.46
Shaanxi	1.32	3.96	7.89	11.90	20.97	0.58	1.35	3.77	11.28	27.15
Gansu	0.20	0.33	1.05	2.90	6.93	0.05	0.14	0.54	2.23	6.10
Qinghai	0.07	0.18	0.29	0.77	1.33	0.03	0.07	0.19	0.65	1.47
Ningxia	0.12	0.19	0.50	0.73	1.13	0.03	0.05	0.21	0.49	1.03
Xinjiang	1.67	2.21	3.73	11.39	33.74	0.39	0.76	1.62	5.63	19.99

**Table 3 ijerph-20-01898-t003:** The results of tourism carbon emissions efficiency (kgCO_2_-e/CNY).

Region	2000	2005	2010	2015	2019	Mean	Rank
Beijing	0.113	0.091	0.055	0.034	0.023	0.061	1
Tianjin	0.312	0.201	0.112	0.069	0.055	0.142	18
Hebei	0.416	0.218	0.169	0.130	0.097	0.193	21
Shanxi	0.373	0.343	0.359	0.188	0.137	0.288	28
Inner Mongolia	0.343	0.476	0.546	0.291	0.201	0.375	29
Liaoning	0.235	0.237	0.213	0.140	0.112	0.186	20
Jilin	0.333	0.326	0.180	0.131	0.106	0.208	23
Heilongjiang	0.334	0.307	0.275	0.253	0.187	0.274	26
Shanghai	0.135	0.132	0.080	0.049	0.034	0.088	6
Jiangsu	0.104	0.091	0.060	0.042	0.035	0.068	2
Zhejiang	0.145	0.118	0.084	0.057	0.035	0.086	5
Anhui	0.139	0.107	0.098	0.092	0.068	0.103	8
Fujian	0.120	0.119	0.082	0.051	0.039	0.086	4
Jiangxi	0.182	0.156	0.108	0.079	0.066	0.123	13
Shandong	0.104	0.232	0.156	0.072	0.059	0.128	14
Henan	0.241	0.184	0.116	0.086	0.061	0.136	16
Hubei	0.190	0.168	0.146	0.073	0.062	0.129	15
Hunan	0.122	0.145	0.110	0.079	0.063	0.103	9
Guangdong	0.137	0.115	0.074	0.051	0.039	0.083	3
Guangxi	0.147	0.160	0.129	0.073	0.056	0.112	10
Hainan	0.175	0.155	0.121	0.092	0.055	0.119	11
Chongqing	0.127	0.128	0.081	0.066	0.044	0.091	7
Sichuan	0.198	0.163	0.118	0.061	0.045	0.120	12
Guizhou	0.941	0.538	0.356	0.237	0.157	0.428	30
Yunnan	0.202	0.214	0.149	0.073	0.046	0.138	17
Shaanxi	0.229	0.294	0.209	0.106	0.077	0.181	19
Gansu	0.385	0.235	0.194	0.130	0.114	0.208	24
Qinghai	0.246	0.267	0.156	0.118	0.090	0.196	22
Ningxia	0.421	0.358	0.242	0.149	0.109	0.282	27
Xinjiang	0.430	0.290	0.231	0.202	0.169	0.259	25
Mean	0.253	0.210	0.167	0.109	0.081	--	--

**Table 4 ijerph-20-01898-t004:** Moran’s I of tourism carbon emissions efficiency from 2000 to 2019.

Year	Moran’ I	*p* Value	Year	Moran’ I	*p* Value
2000	0.1066	0.0613 *	2010	0.2108	0.0017 ***
2001	0.1583	0.0131 **	2011	0.1937	0.0032 ***
2002	0.1541	0.0172 **	2012	0.1721	0.0055 ***
2003	0.1529	0.0168 **	2013	0.1798	0.0077 ***
2004	0.1761	0.0078 ***	2014	0.1816	0.0068 ***
2005	0.2551	0.0003 ***	2015	0.1838	0.0072 ***
2006	0.2407	0.0007 ***	2016	0.1659	0.0146 **
2007	0.2440	0.0005 ***	2017	0.1793	0.0091 ***
2008	0.2254	0.0012 ***	2018	0.1945	0.0051 ***
2009	0.2178	0.0017 ***	2019	0.1979	0.0045 ***

Note: The symbols ***, **, and * denote significance at the 1%, 5%, and 10% levels, respectively. All data in this part retain four decimal places.

**Table 5 ijerph-20-01898-t005:** Detector results for the influencing factors of tourism carbon emissions efficiency.

Period	GRP	UR	MIN	TRE	TTI	ECP	IEP
2005–2019	0.1005 ***	0.0933 ***	0.3230 ***	0.0716 ***	0.1719 ***	0.5081 ***	0.0656 ***
The Eleventh Five-Year Plan	0.1264 ***	0.1164 ***	0.4561 ***	0.2881 ***	0.1582 ***	0.6729 ***	0.1266 ***
The Twelfth Five-Year Plan	0.0999 **	0.1248 ***	0.2960 ***	0.0432	0.1784 ***	0.5583 ***	0.0678 **
The Thirteenth Five-Year Plan	0.2257 ***	0.1584 ***	0.4366 ***	0.1337 ***	0.2083 ***	0.6041 ***	0.1072 **

Note: The symbols ***, ** denote significance at the 1%, 5% levels, respectively. All data in this part retain 4 decimal places.

**Table 6 ijerph-20-01898-t006:** Interaction results for the influencing factors of tourism carbon emissions efficiency.

Factor Interaction Type	The Eleventh Five-Year Plan Period	The Twelfth Five-Year Plan Period	The Thirteenth Five-Year Plan Period
GRP & UR	0.2570	0.1630	0.3142
GRP & MIN	0.6628	0.5605	0.5618
GRP & TRE	0.5570	0.3777	0.3431
GRP & TTI	0.5111	0.5883	0.5852
GRP & ECP	0.8077	0.7929	0.6926
GRP & IEP	0.3645	0.3401	0.3353
UR & MIN	0.5886	0.4854	0.5693
UR & TRE	0.4728	0.3431	0.3323
UR & TTI	0.3701	0.5083	0.6522
UR & ECP	0.8257	0.7641	0.7175
UR & IEP	0.4398	0.4392	0.4575
MIN & TRE	0.5539	0.3446	0.6458
MIN & TTI	0.5398	0.4085	0.6284
MIN & ECP	0.7518	0.7515	0.6627
MIN & IEP	0.5464	0.4612	0.6304
TRE & TTI	0.4315	0.3999	0.6419
TRE & ECP	0.8199	0.7028	0.7344
TRE & IEP	0.4174	0.2161	0.3639
TTI & ECP	0.7047	0.5901	0.7007
TTI & IEP	0.4823	0.5705	0.5898
ECP & IEP	0.7538	0.7977	0.7397

## Data Availability

The data used in this study are available from the corresponding author on reasonable request.

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
