# Peer review of "Spatial-Temporal Variation and Influencing Factors of Regional Tourism Carbon Emission Efficiency in China Based on Calculating Tourism Value Added"

_ijerph, 2023, doi:10.3390/ijerph20031898_

Round 1

Reviewer 1 Report

Dear Authors,

As a general impression after reading the article, your paper excels through the extension and depth of the analyzes that have been presented. But the results obtained are, to some extent, at the level of college students. It is a classical case study based on existing statistical data, in which established indicators are calculated, after which the results are commented.

However, a scientific paper should propose, based on the research carried out, new models, new factors to express the specifics of the destination which was analyzed, or adaptations of the existing models according to the characteristics of the analysed tourist destinations or make comments on the limits of the existing models, etc.

 -       Line 86. In the previous sentence, authors mentioned three groups of factors. After which, a single factor is referred to: Gosling (14,20)..... The meaning is unclear.

-       At page 138 authors mentioned the following: to characterize the economic value of tourism products or services..... In tourism, the analysis of the economic value differs, as manner of expression, from that of other sectors in the economy. This is mainly due to the fact that products and services in tourism are often intangible and therefore, cannot be compared to products from industry, agriculture, etc. Consequently, the economic value or efficiency of carbon emissions are not analyzed at the level of products and services. Instead, the analysis is carried out at the level of different tourism activities, such as accommodation, tourism transport, catering, etc. The same expression formula is found in other sentences, for example on pages 166, 208..

-       Lines 181-182. What authors meant when saying: The parameters used in the tourism carbon emissions estimation model are too old. What are authors referring to?

-        Line 189. The explanations given in the paragraph starting at this line are unclear. What does the term terminal consumption refer to?

-       The phrase from lines 414-415: Refer to the existing research results of the influencing factors of tourism carbon emission efficiency [45,46,47,48] and select the appropriate influencing factors. Next, at lines 418-419 it was mentioned: influencing factor framework of tourism carbon emission efficiency in China and selects the following indicators. How were these factors selected by the authors? The selection process was based on some criteria? If so, which were these criteria?

The conclusion section needs improvements by better highlighting the limits of the research and specific proposals for improving the analysis methods.

The bibliography does not meet the standards of the journal. Some authors are given their first name as in the works cited in no. 21, 36, 40,..

Author Response

Thanks for the suggestions of the reviewer, the authors has revised as follows:

1.Line 86. In the previous sentence, authors mentioned three groups of factors. After which, a single factor is referred to: Gosling (14,20)..... The meaning is unclear.

Previous text expression is unclear, follow-up analysis of the total number of documents to supplement: Chinese scholars ' research on tourism carbon emission efficiency and its related spatial characteristics and influencing factors have been increasing in recent years. For example, Ruan…

  1. Line 138 authors mentioned the following: to characterize the economic value of tourism products or services..... In tourism, the analysis of the economic value differs, as manner of expression, from that of other sectors in the economy. This is mainly due to the fact that products and services in tourism are often intangible and therefore, cannot be compared to products from industry, agriculture, etc. Consequently, the economic value or efficiency of carbon emissions are not analyzed at the level of products and services. Instead, the analysis is carried out at the level of different tourism activities, such as accommodation, tourism transport, catering, etc. The same expression formula is found in other sentences, for example on pages 166, 208..

In order to make the expression more accurate, we have changed "economic benefit and environmental impact " into " economic benefit and negative environmental impact ", and also haved changed "the economic value of tourism products or services " into "the economic value of tourism industry."

3.Lines 181-182. What authors meant when saying: The parameters used in the tourism carbon emissions estimation model are too old. What are authors referring to?

This part of the content overlaps with the review part of the tourism carbon emission measurement method. We have deleted this part from the method description and integrates it into the literature review part after modification. The modified content are as follows : Due to the differences in development of tourism in various regions and the restrictions of research conditions, the carbon dioxide emission factor and energy consumption co-efficient of transportation, accommodation facilities and tourism activities have not been unified[14,39], and some scholars ' research on the carbon emission factor and energy consumption coefficient used in the tourism carbon emission estimation model in recent years is still based on more than ten years ago, which is too old and ignores the changes in tourism caused by economic development and technological innova-tion[7,40]. For example, the carbon emission factors from different travel modes of highway, aviation , railway , other are 1.8 MJ/pkm, 2.0 MJ/pkm, 1.0 MJ/pkm, 0.9 MJ/pkm in Shi et al.’s research, the carbon emission factors from different travel modes of railway, road, waterway are 1 MJ/pkm, 1.8 MJ/pkm, 1.48 MJ/pkm in Wang et al.’ research;. When estimating energy consumption in the accommodation facilities, Gössling et, al. take the average energy consumption per unit to 130 MJ bed-1 night, Shi et, al. take 155 MJ bed-1 night, and Ma et, al. and Wang et, al. and Ma et, al. And Wang et, al. estimated carbon emission factor and energy consumption coefficient of tourism emissions still refer to the research results of Shi et, al. in 2011, not combined with the current development of China to re-estimate the carbon emission factor and energy consumption coefficient.

  1. Line 189. The explanations given in the paragraph starting at this line are unclear. What does the term terminal consumption refer to?

The previous description is not accurate, the terminal consumption refers to the final energy consumption. According to the definition from OECD and IEA,final energy consumption is the amount of energy consumed after subtracting the losses of the three intermediate links of energy processing, conversion, storage and transportation and the energy used by the energy industry.

  1. The phrase from lines 414-415: Refer to the existing research results of the influencing factors of tourism carbon emission efficiency [45,46,47,48] and select the appropriate influencing factors. Next, at lines 418-419 it was mentioned: influencing factor framework of tourism carbon emission efficiency in China and selects the following indicators. How were these factors selected by the authors? The selection process was based on some criteria? If so, which were these criteria?

Considering the difficulty of data acquisition, comprehensiveness of data and limitation of measurement model, it is unlikely to take all influencing factors into consideration. Therefore, this study mainly considers the influencing factors of tourism carbon emission efficiency based on tourism carbon emission efficiency and relevant existing research results, connotation of tourism carbon emission efficiency and measurement model, etc. Select some indicators to explore the intensity of their effect on tourism carbon emission efficiency, and add the explanation of the impact factors in the paper. The selection description of influencing factors is modified as follows: 1) Based on the measurement model of tourism carbon emission efficiency, this study explores the influence mechanism of various factors on tourism carbon emission efficiency from the perspectives of regional development, tourism industry development and resource-environment. 2) Referring to the existing research results on tourism carbon emission efficiency, we select the level of economic development, the level of opening-up, the level of urbanization, the level of opening-up, the level of urbanization, the tourism industry environment, the tourism infrastructure, the level of scientific and technological innovation, and the intensity of environmental regulation.

  1. The bibliography does not meet the standards of the journal. Some authors are given their first name as in the works cited in no. 21, 36, 40,..

All reference formats have been checked and modified.

Reviewer 2 Report

This paper uses Tourism related carbon emission efficiency to analyze the relationship between tourism carbon emission efficiency and sustainable development. The novelty of this article is the introduction of tourism value added, but there are many shortcomings in this article, which I list in order of importance below.

1. First of all, the title does not make it clear what "added value" is. The author should make this clear to avoid misunderstandings.

2. The appearance of "geographic probe" in the keywords seems irrelevant to the title, the author may consider removing it or replacing it with a new keyword.

3. In the introduction of 1, "It also has an impact on the economy" needs to be stated as "negative impact", which is more intuitive.

4. 2 When describing "Methods", the literature review part overlaps with the previous part, so it is necessary to consider which part should be deleted.

5. Part 4 corresponds well to the "3 Methods". However, the influencing factors selected by the author in the estimation of 4.4 Influencing factors are not very convincing, and are somewhat separated from the other parts of "4 Results".

Author Response

Thanks for the suggestions of the reviewer, the authors has revised as follows:

1.First of all, the title does not make it clear what "added value" is. The author should make this clear to avoid misunderstandings.

In order to express the research content more clearly, the topic is changed to ‘Spatial-temporal Variation and Influencing Factors of Regional Tourism Carbon Emission Efficiency in China based on Calculating Tourism Value Added’

2.The appearance of "geographic probe" in the keywords seems irrelevant to the title, the author may consider removing it or replacing it with a new keyword.

In this research,we use geographical detector to analysis the influencing factors of tourism carbon emission efficiency. It is a key part of the article, so we put “geographical detector” in the keywords.

  1. In the introduction of 1, "It also has an impact on the economy" needs to be stated as "negative impact", which is more intuitive.

This part we change it to ‘negative environmental impact’.

  1. 2When describing "Methods", the literature review part overlaps with the previous part, so it is necessary to consider which part should be deleted.

We have deletedThere are many estimation methods of tourism carbon emissions in the existing research results…The specific calculation process is as follows:’ from the method description and integrates The main reasons for the differences are as follows: 1) Due to the differences in devel-opment of tourism in various regions and the restrictions of research conditions, the carbon emission coefficient of transportation, accommodation facilities and tourism activities has not been unified. 2) The tourism industry involves many activity sectors. The estimation results of tour-ism carbon emissions obtained by existing methods are often lower than the actual results, with large leakage’ into the literature review part of the tourism carbon emission measurement method.

  1. Part 4 corresponds well to the "3 Methods". However, the influencing factors selected by the author in the estimation of 4.4 Influencing factors are not very convincing, and are somewhat separated from the other parts of "4 Results".

Considering the difficulty of data acquisition, comprehensiveness of data and limitation of measurement model, it is unlikely to take all influencing factors into consideration. Therefore, this study mainly considers the influencing factors of tourism carbon emission efficiency based on tourism carbon emission efficiency and relevant existing research results, connotation of tourism carbon emission efficiency and measurement model, etc. Select some indicators to explore the intensity of their effect on tourism carbon emission efficiency, and the explanation of the selected influencing factors is added as the basis to enhance the convincing of the selected influencing factors. For example:(1) The level of economic development : the level of economic development affects capital investment, resource consumption, industrial structure... (2) The level of opening-up : the higher the level of opening-up to the outside world, the higher the total amount of foreign investment that the tourism industry can introduce...

4.4 is mainly the analysis of the influencing factors, and it belongs to the empirical part, so we put in parts of ‘4 Results’

Reviewer 3 Report

In this paper, the authors discuss the spatial-temporal characteristics and influencing factors of tourism carbon emission efficiency of China. It is well designed and well written. Here are some suggestions:

1.    Abstract: you should tell the reader the meaning of q value. The meaning of the environment of the tourism industry is not clear. The interaction between the two factors, which two factors should be clearly stated.

2.    Introduction, the websites can be deleted.

3.    Line 86, the meaning of this factor is not clear.

4.    Line 108, Malin[6] should be Song and Li[6],Liu should be Liu et al., Zhang should be Zhang et al.

5.    Reference [34] is related to the Yangtze River Delta, not the Yellow River Basin.

6.    Zeng Guojun should be Zeng and Cai, Li Jiangfan should be Li and Li.

7.    Since you are submitting to an international journal, the wording of domestic and foreign scholars are not proper.

8.    Formula (1), summation symbol was omitted. Formula (4), the subscript I of TCE was omitted.

9.    Line 199, TC is the total tourism carbon emissions in China, TC is the tourism carbon emissions, please make a check.

10.          What does ESDA and LISA mean?

11.          What is the difference between kgCO2-e/CNY and kgCO2-e/¥?

12.          Table 4, please explain more how the tourism resources endowment and transportation infrastructure were measured.

13.          Openness-up should be opening-up or openness.

14.          Line 490, check the brackets.

15.          Line 554, the threshold of q is different with earlier version.

16.          Line 577, the energy consumption structure is optimized should be the energy consumption structure should be optimized.

17.          Line 578, the tourism carbon emission efficiency in regions with high traditional coal energy consumption is higher? Make a check.

18.          Make a careful check of the references.

Author Response

Thanks for the suggestions of the reviewer, the authors has revised as follows:

1.Abstract: you should tell the reader the meaning of q value. The meaning of the environment of the tourism industry is not clear. The interaction between the two factors, which two factors should be clearly stated.

We have added the meaning of q value in the abstract: The q value represents the intensity of the impact factor on tourism carbon emission efficiency.

An introduction to the tourism industry environment was added to the subsequent impact factor indicators. Tourism industry environment : this study uses tourism resource endowment ( TRE ) as a measure of the tourism industry environment to explore its impact on tourism carbon emission efficiency. Tourism resources are a strong support for the development of the tourism industry and the most important factor to attract tourists to the destination. The richer the tourism resources, the better the economic benefits of tourism, the more can provide a guarantee for the improvement of regional tourism ecological level.

The interaction between the two factors means any two of the selected factors, not just two specific factors, The description of the article is not accurate enough, we have changed ‘The interaction between the two factors’ to ‘two-factor interaction’.

  1. Introduction, the websites can be deleted.

The websites have been deleted.

  1. Line 86, the meaning of this factor is not clear.

The original description is not accurate enough, we have changed it as follows: Due to the differences in development of tourism in various regions and the restrictions of research conditions, the carbon emission coefficient of transportation, accommoda-tion facilities and tourism activities has not been unified.

  1. Line 108, Malin[6] should be Song and Li[6],Liu should be Liu et al., Zhang should be Zhang et al.

This part has been amended, the modified content are as follows : Song and Li [6], Liu et al. [18], Zhang et al. [25] and others analyzed ...

  1. Reference [34] is related to the Yangtze River Delta, not the Yellow River Basin.

This part has been amended, the modified content are as follows :For example, Chinese scholars have measured the regional tourism carbon emissions in the Chinese coastal cities [18]and the Yangtze River Economic Belt [33,34].

  1. Zeng Guojun should be Zeng and Cai, Li Jiangfan should be Li and Li.

This part has been amended, the modified content are as follows :Zeng and Cai measured the tourism value added in 30 provincial regions…

This study draws on the idea of the tourism consumption stripping coefficient proposed by Li and Li [41].

  1. Since you are submitting to an international journal, the wording of domestic and foreign scholars are not proper.

We have changed ‘domestic and foreign scholars’ to ‘many scholars,and made a careful check of avoiding that argument.

  1. Formula (1), summation symbol was omitted. Formula (4), the subscript I of TCE was omitted.

Due to our negligence, this part has been amended. We have changed Formula (1) to , and changed ‘TCE’ to ‘TCEi’.

  1. Line 199, TC is the total tourism carbon emissions in China, TC is the tourism carbon emissions, please make a check.

This part has been amended, we have kept ‘TC is the total tourism carbon emissions in China’ in the article and deleted ‘TC is the tourism carbon emissions’ after modification.

  1. What does ESDA and LISA mean?

The full name has been added after amend.

Exploratory Spatial Data Analysis (ESDA)

Local Indicators of Spatial Association(LISA)

  1. What is the difference between kgCO2-e/CNY and kgCO2-e/¥?(?)

The meaning of ‘kgCO2-e/CNY’ and ‘kgCO2-e/¥’ is the same, which has not been unified in the description of the article, it has been uniformly modified to ‘kgCO2-e/CNY’.

  1. Table 4, please explain more how the tourism resources endowment and transportation infrastructure were measured.

After the illustration of the influencing factor indices of tourism carbon emissions efficiency, the measurement process of tourism resources endowment, transportation infrastructure is added. The supplementary contents are as follows:

The tourism resource endowment index refers to the existing research results [49], and combined with the actual differences of different grades of scenic spots and star hotels, assigns five points, three points and one points to the number of AAAAA, AAAA and AAA scenic spots, and generates the index of "tourism scenic spot resources index". The number of five-star hotels, four-star hotels and three-star hotels is assigned according to five points, three points and one points respectively to obtain the "star hotel resource index" index, and then the tourism resource endowment index is summarized.

Tourism transportation infrastructure is weighted and calculated by different levels of road transportation mileage [50], and the specific calculation formula is as follows:

In this study, n is 5, is the speed of roads of type i (120km/h for railways, 100km/h for expressways, 80km/h for primary highways, 60km/h for secondary highways and 40km/h for grade highways), is the length of road transport lines of type i, and S is the area of the province.

  1. Openness-up should be opening-up or openness.

This part has been amended, We have changed ‘Openness-up’ to ‘opening-up’.

  1. Line 490, check the brackets.

This part has been amended, the modified content is as follow: it is manifested as nonlinear enhancement q(Xi∩Xj)>q(Xi)+q(Xj) or double factor en-hancement (q(Xi∩Xj)>max{q(Xi),q(Xj)}.

  1. Line 554, the threshold of q is different with earlier version.

There is a misrepresentation here. It has been changed to ‘According to the q-value, all types of indices are divided into dominant factors (0.5 ≤ q≤ 1), inducing factors (0.2≤q<0.5) and driving factors (0≤q<0.2).

  1. Line 577, the energy consumption structure is optimized should be the energy consumption structure should be optimized.

This part has been amended, We have changed the energy consumption structure is optimized’ to the energy consumption structure should be optimized.

  1. Line 578, the tourism carbon emission efficiency in regions with high traditional coal energy consumption is higher? Make a check.

We searched the relevant data in recent years and found that the data presentation results are consistent with the description of the article. Due to space limitations, only the 2019 results are supplemented in the article, and revises the description of the article as follows: From the regional differences in tourism carbon emission efficiency, it can be found that the values of tourism carbon emission efficiency in regions with high traditional coal energy consumption (The provinces in China with a high proportion of regional energy consumption structure in 2019 include Hebei, Ningxia, Shanxi, Inner Mongolia, Yunnan and Guizhou) is higher than that in other regions.

  1. Make a careful check of the references.

We have made a careful check of all references and modified them.

Round 2

Reviewer 1 Report

Dear Authors,

I consider that the present version of your article meets the requirements for publishing.

There is only a small element I noticed, that needs corrections. At lines 261-262, when identifying the factors chosen in the analysis, 2 out of them, i.e. the level of opening-up and the level of urbanization, were two times mentioned in the same phrase.

At the same time, even if authors did not mention in their reply, I appreciate the fact that the Conclusion section is much improved.

Author Response

Dear reviewer,

Thank you for your review comments.

The following places have been modified:
We have deleted ‘the level of opening-up, the level of urbanization’, which were two times mentioned at lines 261-262.

Kind regards

Reviewer 2 Report

All my concerns have been addressed.

Author Response

Dear reviewer,

Thank you for your review comments in round one. 

Kind regards